# Cost-Effectiveness of Screening to Identify Pre-Diabetes and Diabetes in the Oral Healthcare Setting

**Lan Gao [1,*]**, **Elise Tan [1]**, **Rodrigo Mariño [2]**, **Michelle King [2]**, **Andre Priede [2]**, **Geoff Adams [2]**, **Maria Sicari [2]** and **Marj Moodie [1]**

1    Deakin Health Economics, Institute for Health Transformation, Deakin University, Melbourne 3125, Australia
2    The Melbourne Dental School, The University of Melbourne, Melbourne 3053, Australia
\*    Correspondence: lan.gao@deakin.edu.au; Tel.: +613-9244-5533; Fax: +613-9244-6624

**Abstract:** Background: This study assesses the long-term cost-effectiveness of this screening protocol from a healthcare system perspective. Methods: Australians presenting to private oral healthcare practices recruited to the iDENTify study were included as the study population. A Markov model preceded by a decision tree was developed to assess the intervention's long-term cost-effectiveness when rolled out to all eligible Australians, and measured against 'no-intervention' current practice. The model consisted of four health states: normoglycaemia; pre-diabetes; type 2 diabetes and death. Intervention reach of various levels (10%, 20%, 30%, and 40%) were assessed. The model adopted a 30-year lifetime horizon and a 2020 reference year. Costs and benefits were discounted at 5% per annum. Results: If the intervention reached a minimum of 10% of the target population, over the lifetime time horizon, each screened participant would incur a cost of $38,462 and a gain of 10.564 QALYs, compared to $38,469 and 10.561 QALYs for each participant under current practice. Screening was associated with lower costs and higher benefits (a saving of $8 per person and 0.003 QALYs gained), compared to current standard practice without such screening. Between 8 and 34 type 2 diabetes cases would be avoided per 10,000 patients screened if the intervention were taken up by 10% to 40% of private oral healthcare practices. Sensitivity analyses showed consistent results. Conclusions: Implementing type 2 diabetes screening in the private oral healthcare setting using a simple risk assessment tool was demonstrated to be cost-saving. The wider adoption of such screening is recommended.

**Keywords:** diabetes screening; dental clinic; oral healthcare practices; cost-effectiveness





## 1. Introduction

Diabetes is associated with many oral complications, with diabetic people more likely to experience periodontal problems and achieve poor treatment outcomes, leading to eventual tooth loss, compared to people without diabetes [1]. Consistent with other diabetes related complications, susceptibility to periodontitis increases with sub-optimal glycaemic control [2].

Early diagnosis plays an important role in the prevention and management of T2D. The early stage of T2D is often asymptomatic, and eventual diagnosis is usually delayed by 4 to 8 years from the time of actual onset [3]. This represents a significant missed opportunity to initiate early interventions to halt disease progression, since early identification of high-risk individuals can postpone or even prevent the onset of T2D. Pre-diabetes is a reversible condition associated with an increased risk of cardiovascular disease, coronary heart disease, stroke and all-cause mortality and if left untreated, 15% to 30% of people with pre-diabetes will progress to T2D within 5 years [4].

The oral healthcare setting provides a good location for opportunistic screening of individuals with undiagnosed medical issues, as oral healthcare professionals are likely to encounter asymptomatic patients with undiagnosed pre-diabetes or T2D. A systematic

review synthesised evidence relating to the role of oral healthcare teams in identifying individuals with undiagnosed pre-diabetes or T2D in oral healthcare settings and concluded that there may be benefits for engagement of the oral health workforce to identify people with undiagnosed pre-diabetes and T2D, while high-quality research was needed to demonstrate the clinical and cost-effectiveness of such a practice [5].

The iDENTify study developed and evaluated an innovative approach for the identification of pre-diabetes and T2D in the private oral health setting [6]. iDENTify, developed in alignment with Diabetes Australia's National Diabetes Strategy and Action Plan, aimed to test the effectiveness of T2D risk assessment using The Australian Type 2 Diabetes Risk Assessment Tool (AUSDRISK) [7] within the private oral healthcare setting followed by referral to a General Practitioner (GP) for full diagnosis and management. Participating oral health patients identified as at risk of pre-diabetes and T2D were advised of this and offered a referral to a GP for further assessment and diagnosis. Based on a mixed-study design, it was found that the screening protocol was well accepted by patients and oral health professionals, and the procedures were easy to implement and could be incorporated into routine oral healthcare practice [8]. However, the long-term economic credentials of the screening protocol remain unknown. This study sets out to model the lifetime cost-effectiveness of the iDENTify screening protocol from an Australian healthcare system perspective.

## 2. Materials and Methods

### 2.1. The Intervention

The Australian Type 2 Diabetes Risk Assessment Tool (AUSDRISK) identifies patients at high risk of developing type 2 diabetes and consists of 10 items that assess risk factors including age, gender, country of birth, family history of diabetes, history of high blood glucose, hypertension, smoking status, fruit and vegetable intake, physical activity levels and waist circumference. Scores range from 0 to 38 and reflect the probability of developing diabetes within the next 5 year [9]. The usual cut-off of 6 is associated with a sensitivity of 97.7% (95% CI 95.4–99.0) and specificity of 20.0% (95% CI 18.9–21.0) [10] in identifying high risk people.

The iDENTify protocol involved screening for pre-diabetes or T2D with the AUSDRISK in the private oral healthcare setting and referral of patients at an elevated risk of having or developing pre-diabetes or T2D to a GP by their oral health professionals. It recruited a convenient sample consecutively. Patients aged 35 or over, without a previous diagnosis of pre-diabetes/T2D were eligible to participate in the trial. Upon completion of the AUSDRISK tool, all patients received, from the oral health professional appropriate health advice (healthy food intake and physical activity) and a periodontal assessment (based on the Community Periodontal Index) [11]. Those considered intermediate or high risk (i.e., AUDSRISK score of 6 or greater) were referred to their GP for further assessment and management. Referred patients were provided with a GP referral pack which included a personalised referral letter, assessment proforma and a study information brochure for the GP. GPs were asked to continue with their usual clinical practice for the management of T2D, and to return the diabetes assessment results to the referring oral healthcare professionals. The study was approved by the University of Melbourne Human Research Ethics Committee (Ethics ID: 1749595), and all the participants provided written informed consent. The comparator was usual or 'standard' care in the private oral healthcare setting in which screening for T2D is not available.

### 2.2. Study Population

Australians presenting to private oral health practices that had been recruited into the iDENTify study were included as the study population. Briefly, a total of 51 private oral healthcare practices and 76 oral health practitioners in both metropolitan and rural Victoria participated in the study. In total, 806 patients were screened for T2D in the oral healthcare setting; after four persons were excluded because of age restrictions, and one

chose to withdraw, the final sample was 801. The modelled population was characterised according to the high-risk population recruited in the primary study.

### 2.3. Structure of the Simulation Model

A Markov model preceded by a decision tree was developed to assess the long-term cost-effectiveness of the iDENTify intervention if it was extrapolated to private dental clinics throughout Australia. The model structure was similar to published economic evaluations in T2D screening [12]. This model aimed to quantify the benefits of early identification of individuals with undiagnosed pre-diabetes/T2D at baseline, compared to no intervention in the dental setting. The economic modelling set to ascertain the benefits from one-off screening using iDENTIFY (i.e., new people identified by such screening in later years are not considered). Intervention reach of various levels (10%; 20%; 30% and 40%) was assessed within the intervention cohort. Data such as the proportion of patients in the intervention group identified as pre-diabetes/T2D and the costs of the intervention were drawn from the study and used in combination with other model inputs sourced from published literature.

The model adopted a 30-year lifetime horizon, with annual cycles and was built in TreeAge Pro 2019, R2 (TreeAge Software Inc., Williamston, MA, USA). Costs and effectiveness were discounted at a rate of 5% [13].

Figure 1A,B illustrate the model structure.

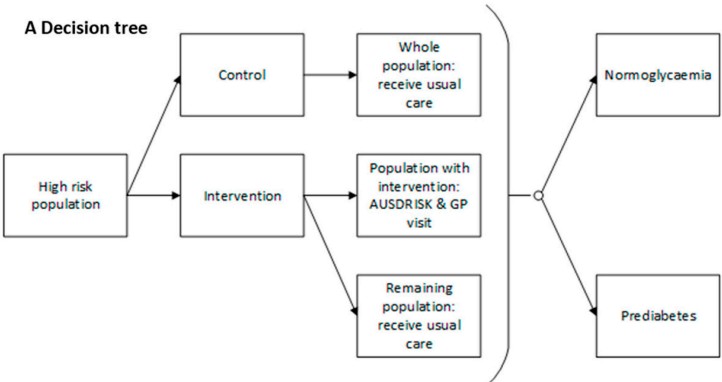

**A Decision tree**

Note: Comparison between the intervention and control cohorts. Within the intervention cohort, intervention reach may differ, that is the proportion of population receiving the intervention can vary. Initial health states are normoglycaemia and pre-diabetes.

**B Markov state transitions**

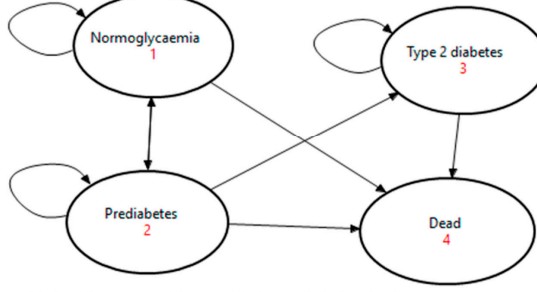

Note: The possible health state transitions within the modelled evaluation.

**Figure 1.** Model structure (**A**) of the decision tree component and (**B**) of the Markov component) for assessing the cost-effectiveness.

### 2.4. Model Inputs

#### 2.4.1. Transition Probabilities

The probability of pre-diabetes being identified by the intervention was based on study data. This was compared to the prevalence in Australia of pre-diabetes in the high risk population in the control arm (using the health economics model) and within the intervention arm to account for unidentified individuals with pre-diabetes in the study.

The remaining participants were assumed to be normoglycaemic (as no patients with undiagnosed T2D were identified in the study). The proportion of people being identified as prediabtes by the intervention had a higher probability of returning to normaoglycaemia due to the lifestyle intervention [14].

Background mortality rate was calculated using age-dependent death rates in Australia for the period 2016–2018 [15]. Increased mortality associated with pre-diabetes/T2D was applied in the model [16]. Except for the proportion of pre-diabetes people being identified earlier, all the other transition probabilities were the same between the iDENTify and usual care arms.

The model assumed all patients identified as intermediate or high risk based on the AUDSRISK tool in iDENTify had one GP visit and subsequently underwent an oral glucose tolerance test for T2D. In terms of pre-diabetes management by GPs, a pragmatic lifestyle intervention based on six sessions with health professionals was assumed.

### 2.4.2. Costs

The model adopted a healthcare perspective. Costs included were: the iDENTify intervention delivery costs (oral health practitioner time, administrative time, consumables and training costs); a GP visit for patients identified as pre-diabetic or with undiagnosed T2D [17]; oral glucose tolerance test for referred patients [18]; a pragmatic lifestyle intervention based on the Life! Diabetes prevention program) and the annual direct medical (medical services, hospitalisation, medications and consumables) and non-medical costs(transport to hospital, supported accommodation, community care services, and special foods) of the various health states [19,20]. The management cost of all health states was informed by a study using Medicare data which captured all medications and healthcare service usage by all Australians. We assumed zero cost in the current practice in prediabetes/T2DM diagnosis. In the standard of care, people with a risk of diabetes would have been picked up via usual healthcare-seeking behaviours. The probability of people being diagnosed in the intervention and control groups would be identical. Therefore, even if there were costs of the current practice, they would be offset in the calculation of incremental cost. Costs were valued in 2020 Australian dollars (1 AUD = USD 0.70).

### 2.4.3. Utility

Effectiveness was measured using life years (in the cost-effectiveness analysis) and quality-adjusted life years (QALYs) (in the cost-utility analysis). QALYs are calculated by the number of years lived multiplied by the utility score for being in particular health state(s). The utility of the various health states used in the model, scored between 0 (dead)–1 (perfect health), were valued based on the EuroQol EQ-5D instrument [21].

The age-dependent utility for T2D was informed by a published study [22]. This was compared to general population norms for the corresponding age group for normoglycaemia [23], whilst the utility for pre-diabetes was calculated using the comparison of pre-diabetes and normal glucose tolerance utilities observed by Makrilakis et al. [24].

All the model inputs are summarised in Table 1.

**Table 1.** Model inputs for the long-term cost-effectiveness analysis.

| Parameter | Value | Range for Sensitivity | Distribution |
|---|---|---|---|
| *Probabilities* | | | |
| Proportion of patients with pre-diabetes in the high risk population, identified via intervention | 0.0625 | | |
| Proportion of patients with pre-diabetes in the high risk population, according to literature | 0.212 [12] | (0.152–0.232) | |
| Transition probability for normoglycaemia to pre-diabetes (no treatment) | 0.05065 [13] | | |

**Table 1.** *Cont.*

| Parameter | Value | Range for Sensitivity | Distribution |
|---|---|---|---|
| Transition probability for pre-diabetes to normoglycaemia (no treatment) | 0.08969 [14,15] | | |
| Transition probability for pre-diabetes to diabetes (no treatment) | 0.11 [16] | (0.098–0.123) | Beta (alpha: 88.89, beta: 719.2) |
| Relative risk for the transition of pre-diabetes to normoglycaemia (due to lifestyle changes) | 1.4 [15,17] | | |
| Relative risk for the transition of pre-diabetes to diabetes (due to lifestyle changes) | 0.74 [18] | (0.58, 0.93) | Gamma (alpha: 100, lambda: 135.14) |
| Relative risk of mortality for pre-diabetes | 2.32 [20] | (1.24–3.40) | |
| Relative risk of mortality for type 2 diabetes | 3.45 [20] | (2.02–4.87) | Gamma (alpha: 100, lambda: 28.986) |
| *Costs ($)* | | | |
| Implementing the intervention per high risk patient identified | $60 | | Gamma (alpha: 100, lambda: 1.674) |
| General practitioner visit | $38.75 [21] | | |
| Oral glucose tolerance test | $18.95 [22] | | |
| Pragmatic lifestyle intervention per high risk patient identified | $433.85 [24] | | |
| Annual direct medical cost per person with normoglycaemia | $2635 [23] | | |
| Annual direct medical cost per person with pre-diabetes | $2875 [23] | | |
| Annual direct medical cost per person with type 2 diabetes | $6091 [23] | | Gamma (alpha: 100, lambda: 0.0164) |
| *Utilities* | | | |
| Normoglycaemia health state | 0.89 [25] | | |
| Pre-diabetes health state | 0.88 [26] | | |
| Type 2 diabetes health state | 0.78 [27] | SD:0.25 | Beta (alpha: 21.22, beta: 5.985) |

### 2.5. Cost-Effectiveness Analysis

Since the Markov cohort model employed parameters in which uncertainty was not evaluated at the individual level, no *p*-values can be supplied for the long-term modelling. Cost-effectiveness of the iDENTify screening protocol was assessed against current practice where such screening protocol does not exist. An incremental cost-effectiveness ratio (ICER) was calculated as the ratio between incremental cost and incremental benefit (i.e., cost per QALY gained). Both costs and QALYs were discounted at 5% per annum. The often-quoted willingness-to-pay per QALY threshold of the Australian dollar (A$) 50,000 was adopted to determine the cost-effectiveness of iDENTify screening protocol [28].

### 2.6. Sensitivity Analysis

Both deterministic and probabilistic sensitivity analyses were undertaken to examine the robustness of base case results. A series of one-way deterministic sensitivity analyses by varying the key model inputs (transition probabilities, costs, and relative risk for the intervention effect) within a plausible range were run and plotted in a Tornado diagram. Distributions of key model variables identified from the deterministic sensitivity analysis were constructed and tested in the probabilistic sensitivity analysis. A cost-effectiveness plane was used to visually present the results from probabilistic sensitivity analysis.

The modelled cost-effectiveness analysis was carried out in accordance with the CHEERS checklist for reporting economic evaluation [27].

Expanded methods are provided in Online document (Supplementary Materials).

## 3. Results

### 3.1. Study Population

A total of 51 private oral healthcare practices and 76 oral health practitioners in both metropolitan and rural Victoria participated in the study. Study participants were recruited over two waves between 2018 and 2020, involving 15 practices in the first wave and 36 in the second wave with five practices participating in both waves. Of the participating oral healthcare practices, 34 were located in the metropolitan Melbourne area and 17 were from rural Victoria. Detailed results were reported elsewhere [6].

The characteristics of the iDENTify participants are summarised in Table 2. In total, 806 patients were screened for T2D in the oral healthcare setting; after four persons were excluded because of age restrictions, and one chose to withdraw, the final sample of 801 patients was included in the analysis. There is no control group in the study.

**Table 2.** Characteristics of the iDENTify study cohorts.

|  | **Participants** | **%** |
|---|---|---|
| **Wave** | | |
| 1 | 305 | 38.1 |
| 2 | 496 | 61.9 |
| **Participant Location** | | |
| Metropolitan | 576 | 72.0 |
| Rural | 225 | 28.0 |
| **Sex** | | |
| Female | 491 | 61.4 |
| Male | 309 | 38.6 |
| **Age Group** | | |
| 34–44 years | 150 | 18.7 |
| 45–54 | 207 | 25.8 |
| 55–64 | 200 | 25.0 |
| 65–74 | 168 | 21.0 |
| 75 and more | 76 | 9.5 |
| **Total** | 801 | 100.0 |

Among the total patients, 104 (12.7%) were classified as low risk group, 329 (41.6%) were in the intermediate risk, and 368 (45.7%) were in the high-risk group for T2D. Of the 697 patients as screened intermediate or high risk, 384 (55.1%) were referred on to their GP for further examination (N = 313 patients opted out for further referral). Out of 384 participants referred to the GP, a total of 96 results were returned to oral healthcare practices and six patients were diagnosed with pre-diabetes; no one was diagnosed with T2D. The characteristics of the modelled population were defined as consistent with those of the high-risk participants (N = 368) from iDENTify. The mean age of the high-risk population was 63 years.

### 3.2. Modelled Cost-Effectiveness Analysis

There are an estimated 1701 dentist clinics and 5757 oral health practitioners registered in Victoria [25,26]. The iDENTify trial included approximately 3% and 1% of clinics and

practitioners, respectively. To assess the long-term economic credentials of iDENTify at a larger scale, our study examined an intervention reach of various levels, from 10–40% across Victoria.

If the iDENTify intervention could reach 10% of the target population, over the lifetime time horizon, each screened participant would incur a cost of $38,462, and gains of 10.564 QALYs, compared to costs of $38,469 and 10.561 QALY gains amongst their current practice counterparts (Table 3).

**Table 3.** Results of base case cost-effectiveness analysis.

| | Total Cost | Medical Cost | Non-Medical | Total QALY | Number of T2D ^ |
|---|---|---|---|---|---|
| **Current practice** | $38,469 * | $28,687 | $9783 | 10.561 | 3697 |
| *10% of intervention reach* | | | | | |
| **iDENTify** | $38,462 | $28,686 | $9776 | 10.564 | 3689 |
| **Difference** | −$7.9 | −$1.1 | −$6.8 | 0.003 | 8 |
| **ICER** | | | | Dominant | |
| *20% of intervention reach* | | | | | |
| **iDENTify** | $38,454 | $28,684 | $9769 | 10.567 | 3680 |
| **Difference** | −$15.7 | −$2.2 | −$13.5 | 0.005 | 17 |
| **ICER** | | | | Dominant | |
| *30% of intervention reach* | | | | | |
| **iDENTify** | $38,446 | $28,683 | $9762 | 10.569 | 3672 |
| **Difference** | −$23.6 | −$3.3 | −$20.3 | 0.009 | 25 |
| **ICER** | | | | Dominant | |
| *40% of intervention reach* | | | | | |
| **iDENTify** | $38,438 | $28,682 | $9756 | 10.572 | 3663 |
| **Difference** | −$28.3 | −$1.2 | −$27.1 | 0.011 | 34 |
| **ICER** | | | | Dominant | |

Abbreviations: QALY: quality-adjusted life year; T2D: type 2 diabetes; ICER: incremental cost-effectiveness ratio. Dominant refers to the less costs and more benefits. * 1 AUD = USD 0.70; ^ per 10,000 people.

### 3.3. Sensitivity Analysis

One-way deterministic sensitivity analyses identified that the relative risk for mortality (T2D vs. no T2D), cost of management for T2D, preventive intervention effect (i.e., relative risk of intervention for progression from pre-diabetes to T2D), and utility weight of having T2D were key drivers of the base case ICER (Figure 2).

The probabilistic sensitivity analysis showed that 97.7% of results indicated that iDENTify was less costly and more effective (i.e., dominant) than the no screening comparator; the remaining 2.3% of results, whilst not cost-saving, were cost-effective (Figure 3).

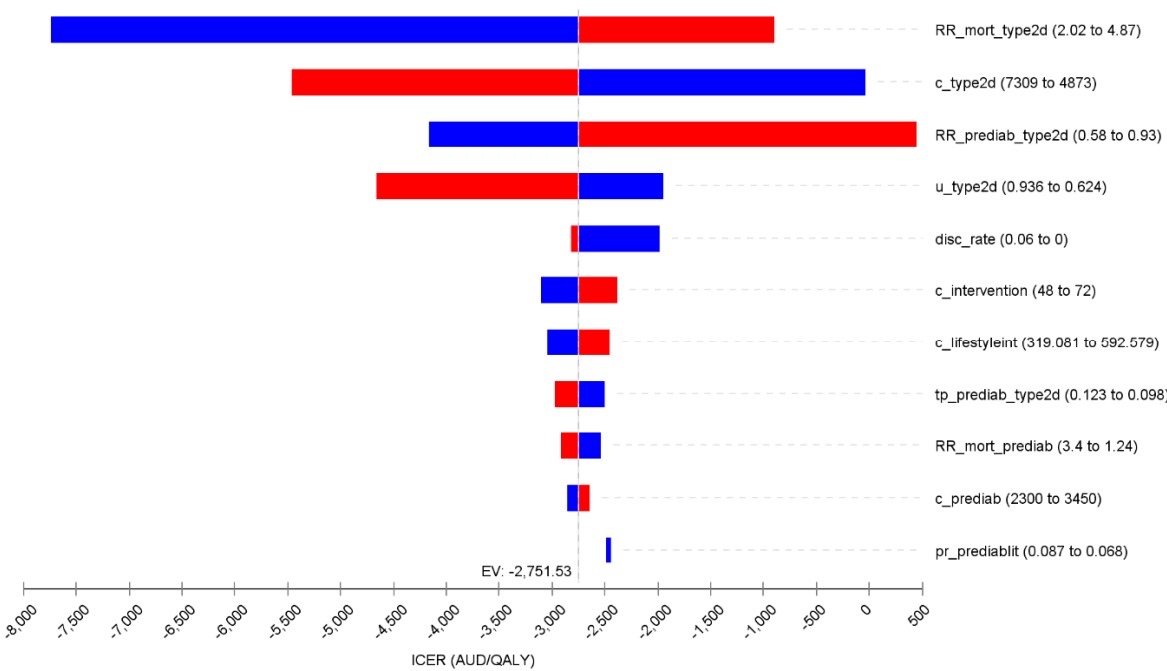

**Figure 2.** Tornado diagram for the one-way deterministic sensitivity analyses. Note: blue bar represents the value of the variable decreases from the base case; red bar denotes the value of the variable increases from the base case. Abbreviations: RR_mort_type2d: Relative risk of mortality in type 2 diabetes; c_type2d: cost of management post type 2 diabetes; u_type2d: utility weights of type 2 diabetes; disc_Rate: discount rate; c_intervention: cost of Identify intervention; c_lifestyleint: cost of lifestyle intervention for prediabetes;tp_prediab_type2d: transition probabili from prediabetes to type 2 diabetes; RR_mort_prediab: Relative risk of mortality in prediabetes; c_prediab: cost management for prediabetes; pr_prediabit: The proportion of prediabetes in "high risk" population as identified in the intervention.

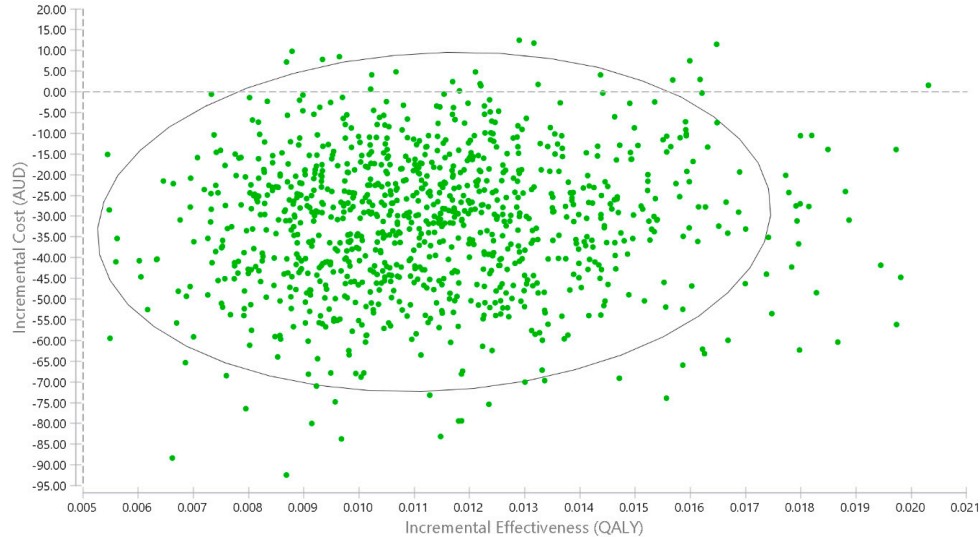

**Figure 3.** Cost-effectiveness plane for the probabilistic sensitivity analysis. Note: 97.7% of results indicating iDENTify is less costly and more effective. The remaining 2.3% of iterations were cost-effective, but not cost-saving. So there is no red dots to show the cost-ineffective results. The eclipse represents the 95% confidence interval of the ICER.

## 4. Discussion

Our study is one of a few to evaluate the cost-effectiveness of a T2D screening protocol implemented in the oral healthcare setting [29]. Even though only 6 out of 96 (6.25%) patients who were referred onto the GP and had their T2D assessment results returned to the oral health practitioner, were diagnosed with pre-diabetes, early intervention with this small proportion of patients produced significant long-term benefits. The small benefits reflect the average across all the screened people who may not have pre-diabetes or T2D. The effectiveness of intervening early in people with pre-diabetes is well established, including lifestyle modifications (i.e., weight loss and physical activity promotion) and pharmacological interventions (i.e., metformin, acarbose, etc.). Through such management activities, the risk of developing T2D could be reduced by between 31% to 58% when compared with the control group [30]. Given such relatively inexpensive and effective pre-diabetes management compared to costly treatment for late-stage T2D, identifying this small proportion of people with pre-diabetes using the iDENTify screening protocol potentially translates to the small long-term cost-saving as modelled in our study.

Studies have been conducted to assess pre-diabetes screening by dentists in the USA. The Dental Practice-Based Research Network study explored the feasibility of random plasma glucose levels for screening for pre-diabetes or previously undiagnosed diabetes in community oral healthcare practices and demonstrated the practicability of such screening in the community [31,32]. A Columbia University study which was conducted in the hospital-based dental practice setting identified 73% of true cases of T2D and pre-T2D using a prediction model [33]. Another US-based study conducted across eleven private oral healthcare practices (n = 816) and a community health centre (n = 206) found that approximately 416 (40.7%) patients screened had an abnormal HbA1c level (i.e., $\geq$5.7%) and were thus referred for diagnosis [34]. All these studies demonstrated that this form of screening was acceptable to the participating oral healthcare professionals, doctors and patients. The results from iDENTify screening protocol are considered comparable and given the unavailability of HbA1c testing devices in oral healthcare settings, its wider implementation is likely to be limited.

The cost-effectiveness of the iDENTify screening protocol was highly dependent on the effectiveness of any preventive intervention for pre-diabetes. Our modelling only incorporated a lifestyle intervention with modest effectiveness t (i.e., RR = 0.74), rather than more potent interventions like acarbose (RR = 0.64) or intensive lifestyle modification (RR = 0.42) [35–37]. As demonstrated in the sensitivity analysis, the effectiveness of the preventive intervention was a key driver of the base case ICER. It is anticipated that adopting more potent interventions in the long-term modelling would potentially achieve a more favourable economic outcome.

There are several limitations worth mentioning. First, the long-term modelling did not simulate the particular type of diabetes-related complications, thus the costs and utility decrement associated with these events are not incorporated. However, this approach is considered not to favour the iDENTify screening protocol. Second, the pre-diabetes detection rate of the iDENTify screening protocol was calculated based on people who had their diabetes testing results returned to the oral health professional (N = 96) since it is difficult to ascertain the pre-diabetes/T2D status of those who had not provided the testing results. The uncertainty from this was tested in the sensitivity analysis. Thirdly, the sensitivity and specificity of AUSTRISK were not considered. However, as a screening tool, the costs associated with false-positive or negatives would have been minimal since it is improbable that any mistreatment or delays in treatment would have been caused.

## 5. Conclusions

Implementing diabetes screening by oral health professionals with a simple risk assessment tool was demonstrated to be cost-saving and associated with greater health benefits. The wider adoption of such screening is recommended.

**Supplementary Materials:** The following supporting information can be downloaded at: https://www.mdpi.com/article/10.3390/endocrines3040062/s1, Expanded methods are provided in Online document [9,10,15,16,38–43].

**Author Contributions:** L.G., E.T., R.M. and M.M. conceived and designed the study; R.M., M.K., A.P., G.A. and M.S. had major role in data collection; L.G. and E.T. undertook the analysis; L.G., E.T., R.M. and M.M. interpreted the results; L.G. drafted the manuscript and E.T., R.M., M.K., A.P., G.A., M.S. and M.M. provided major intellectual inputs to the manuscript. All authors have read and agreed to the published version of the manuscript.

**Funding:** The study was funded by Colgate-Palmolive Pty Limited Australia.

**Institutional Review Board Statement:** The study was approved by the University of Melbourne Human Research Ethics Committee (Ethics ID: 1749595).

**Informed Consent Statement:** Informed consent was obtained from all subjects involved in the study.

**Data Availability Statement:** The datasets used and/or analysed during the current study are available from the corresponding author on reasonable request.

**Conflicts of Interest:** The authors declare no conflict of interest.

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
