# Peer review of "Cost-Effectiveness of Screening to Identify Pre-Diabetes and Diabetes in the Oral Healthcare Setting"

_endocrines, doi:10.3390/endocrines3040062_

Round 1

Reviewer 1 Report

see attached file

Author Response

Thank you very much for your comments. Please find our response in the attached document.

Reviewer 2 Report

This research investigated the effect of oral health screening regarding costs on detecting dysglycemia persons. The issue seems to be important. There are several considerations about scientific methods as below. 

-          I recommend you write full name with an abbreviation at the start of a word in the entire manuscript to better understand.

-          Please explain more detail about the AUDSRISK study. I have no idea why authors performed screening diabetes in oral health setting, because you didn't use dental data (as you mentioned you did periodontal assessment) for screening diabetes.

-          In methods, you mentioned that you calculated medical and non-medical costs, but you didn’t present respectively. So, it is hard to understand the result.

-          The order of the tables is incorrectly marked.

-          Please clarify the fig 1, a decision tree model including probability, outcome, terminal node, chance node, and decision node to estimate the impact on cost-effectiveness.

-          Is this including oral medical expenditures in this cost analysis? It is necessary to clearly define not only the indication, frequency, and interval of medical treatment currently being evaluated, but also what input factors (people, resources) are needed to implement the medical treatment, and whether there are other accompanying medical treatment to be implemented together.

-          What criteria is the intervention reach of various levels (10%; 20%; 30% and 40%)?

Author Response

(The authors gave the same response as above.)

Round 2

Reviewer 1 Report

The revisions and responses address my comment well. No further comment from me.

Reviewer 2 Report

None.